# Genomic Biomarker Discovery in Disease Progression and Therapy Response in Bladder Cancer Utilizing Machine Learning

**DOI:** 10.3390/cancers15194801

**Published:** 2023-09-29

**Authors:** Konstantinos Christos Liosis, Ahmed Al Marouf, Jon G. Rokne, Sunita Ghosh, Tarek A. Bismar, Reda Alhajj

**Affiliations:** 1Department of Computer Science, University of Calgary, Calgary, AB T2N 1N4, Canada; konstantinos.liosis@ucalgary.ca (K.C.L.); rokne@ucalgary.ca (J.G.R.); alhajj@ucalgary.ca (R.A.); 2Memorial Sloan Kettering Cancer Center, New York, NY 10065, USA; 3Department of Medical Oncology, Faculty of Medicine and Dentistry, University of Alberta, Edmonton, AB T6G 2R7, Canada; sunita.ghosh@albertahealthservices.ca; 4Departments of Mathematical and Statistical Sciences, University of Alberta, Edmonton, AB T6G 2J5, Canada; 5Department of Pathology and Laboratory Medicine, Cumming School of Medicine, University of Calgary, Calgary, AB T2N 4N1, Canada; tabismar@ucalgary.ca; 6Departments of Oncology, Biochemistry and Molecular Biology, Cumming School of Medicine, Calgary, AB T2N 4N1, Canada; 7Tom Baker Cancer Center, Arnie Charbonneau Cancer Institute, Calgary, AB T2N 4N1, Canada; 8Prostate Cancer Center, Calgary, AB T2V 1P9, Canada; 9Department of Computer Engineering, Istanbul Medipol University, Istanbul 34810, Turkey; 10Department of Heath Informatics, University of Southern Denmark, 5230 Odense, Denmark

**Keywords:** genomic biomarker discovery, bladder cancer, bioinformatics analysis, elastic-net, therapy response, disease progression

## Abstract

**Simple Summary:**

Cancer in all its forms of expression is a major cause of death. The bladder cancer is also causes the same. finding the biomarkers responsible for the cancer is a challenging task and in certain cases, such as disease progression and therapy response, it become more challenging. The advancements in technology provides latest machine learning methods that help to identify the genomic biomarkers computationally. In this paper, the genomic biomarkers are tracked for bladder cancer from Univeristy of Calgary cohort and different bioinformatics methods, such as differential gene expression, survival rate estimation, consensus gene selection approaches were optimally used. The elastic-net based regression method has been utilized as a machine learning method which shows satisfactory results.

**Abstract:**

Cancer in all its forms of expression is a major cause of death. To identify the genomic reason behind cancer, discovery of biomarkers is needed. In this paper, genomic data of bladder cancer are examined for the purpose of biomarker discovery. Genomic biomarkers are indicators stemming from the study of the genome, either at a very low level based on the genome sequence itself, or more abstractly such as measuring the level of gene expression for different disease groups. The latter method is pivotal for this work, since the available datasets consist of RNA sequencing data, transformed to gene expression levels, as well as data on a multitude of clinical indicators. Based on this, various methods are utilized such as statistical modeling via logistic regression and regularization techniques (elastic-net), clustering, survival analysis through Kaplan–Meier curves, and heatmaps for the experiments leading to biomarker discovery. The experiments have led to the discovery of two gene signatures capable of predicting therapy response and disease progression with considerable accuracy for bladder cancer patients which correlates well with clinical indicators such as Therapy Response and T-Stage at surgery with Disease Progression in a time-to-event manner.

## 1. Introduction

Cancer is a severe disease, often grievously impacting individuals around the world. It is an unexpected and potentially deadly mutation in the normal human cell life cycle, which causes defective or damaged cells to grow uncontrollably either in the organ from which the cells originated or in other parts of the body where they do not belong organically. Such growths, if they turn out to be malignant instead of benign, form tumors that can in turn export cancerous cells to other parts of the body where they form metastases. According to the International Agency for Research on Cancer, a total of almost 19.3 M new cases of cancer were recorded in 2020, and almost 10 M deaths [1].

Being able to harness computational power for bioinformatics in general and specifically for cancer research is both an opportunity and a privilege, considering the scale of this disease and the potential impact a breakthrough in managing the disease might have on the affected population.

Cancer of the bladder (also known as bladder carcinoma) is the fourth most common type of cancer among men and the ninth most common cancer among women in the United States [2], whereas in Canada it ranks the same among men and is the tenth most common type among women [3]. The American Cancer Society for bladder cancer incidence in 2021 predicts that there will be approximately 83,730 new cases of the disease, with about 17,200 of them resulting in death [4]. This type of cancer occurs mainly in older individuals, with the average age at the time of diagnosis being 73. However, especially for men, this disease poses a serious threat, with 1 in 27 males developing it at some time during their life.

In its most common form, bladder cancer is identified as urothelial carcinoma (UC), with more than 90% being classified as such [2,5]. This cancer starts developing in the urothelial cells that constitute the inner layer of the bladder. This type of cell can also be found lining other organs of the urinary tract, such as the urethra or part of the kidneys, making them susceptible to tumor development as well. Aside from this prevalent bladder cancer subtype (which is also dominant in the respective dataset used in this work), there are other types of bladder cancer such as squamous cell carcinoma, adenocarcinoma, small cell carcinoma, and sarcoma. Based on the type of tumor growth, bladder carcinomas can also be grouped into flat and papillary, with the latter often growing towards the hollow center of the bladder while the former does not. Lastly, the carcinoma can also be classified based on how invasive the tumor is. Non-invasive cancers limit their growth to the inner layer of the bladder (the transitional epithelium), while invasive cancers grow into deeper layers of the bladder wall, severely increasing the possibility of further spreading and metastasis. The anatomy of the urinary tract on the left and the enlarged part of bladder on the right. The layers of the bladder wall consisting of transitional epithelium, connective tissue, muscles, and fatty layers are noted. Usually, the papillary and flat tumors are found on the transitional epithelium layer.

During the last fifteen to twenty years, there have been major advancements in genomic profiling technologies, huge leaps in genomic sequencing, and consequently an emergence of selective molecular targeted therapies. These factors have lead to a dramatic increase in the importance of biomarkers where biomarkers are defined as follows: “A biomarker is any substance, structure or process that can be measured in the body or its products and influence or predict the incidence of outcome or disease” [6]. When focusing on cancer research, and with clinical utility in mind, a cancer biomarker can either be considered to be a way of measuring the risk of cancer development, the progression stemming from a tumorous tissue, or even the potential response to a certain type of therapy [7]. Biomarkers can, however, not only be an insightful, decision-making tool for the clinician, but they are also increasingly appearing as links to various abnormalities that are observed in molecular pathways. This fact justifies the application of specific, oftentimes novel, therapeutic (invasive or not) strategies.

This paper aims to find the most relevant qualified genes for therapy response and disease progression among the cohort of bladder cancer patients. With the motivation to identify those genes which may impact highly in these two different scenarios, the research contributions of this paper are indicated below.

A novel method to identify the qualified genes in bladder cancer among the therapy response and disease progression patients.A novel algorithm to separate the histology-based samples grouping into neuroendocrine, utothelial, sqaumous, micropapillary, and other secondary histologies.A pipeline has been designed including differential gene expression (DGE) analysis, Wilcoxon rank sum test, Kaplan–Meier estimation, PCA-based dimensionality reduction and clustering, consensus gene selection approaches.Elastic-net-based regularized regression method has been utilized as a machine learning method with k-fold cross-validation and ROC curve;, time-to-event prediction is used to showcase the prediction result.

## 2. Related Works

The research area of biomarker discovery has made dramatic progress since the beginning of the current century fueled mostly by major technological advancements such as digital computers. These technological advancements have enabled researchers to incorporate computer-implemented computational methods in the discovery process, since they allow for a virtually unlimited number of experiments/simulations, ease of hypothesis testing, and the capability to manipulate vast amounts of data.

Specifically, from a brief review of the available literature for cancer biomarker discovery, one can identify a variety of computational methods employed for that purpose, spanning from traditional statistical approaches to state-of-the-art machine learning techniques and tools tailored to the exact needs of the problem (e.g., pathway analysis).

Before going into the various works that have been conducted and the methods that have been explored, it is interesting to mention a discussion paper from Steyerberg et al. [8] that looks into machine learning techniques and regression methods for risk prediction, in a more or less comparative manner. They point out a particular characteristic of medical data which is that it has a poor signal-to-noise ratio which makes it difficult to develop predictive models. Given the extensive data needs of ML techniques such as for neural networks, this has serious implications for the necessary size of the dataset. They state that penalizations (such as elastic-net) generally result in more parsimonious model specifications that, in turn, are suitable for the nature of such data, and that there is no need for capturing extreme nonlinear relationships in this case. Their conclusion is that they propose to support logistic regression as the “default modeling approach” for medical risk prediction not only in the present, but in the future as well. Such a statement further reinforces the decision to mostly rely on logistic regression models for predictive modeling purposes within the scope of this work.

Despite machine learning methods having matured to their current state, one can find many research works that are mostly, if not exclusively, reliant on more traditional statistical approaches. Specifically, in 2020 Grivas et al. [9] used Cox proportional hazards to correlate neuroendocrine-like bladder cancer subtypes to specific time-to-event endpoints (cancer specific mortality, overall mortality), and logistic regression for other clinical endpoints of a binary nature. Aforementioned statistical analyses were completed using R. For the classification of patients based on said subtype, however, they used a random forest model, as was described in Da Costa et al. [10].

In the same year, Font et al. [11] used Kaplan–Meier curves to derive progression-free survival and disease-specific survival estimations among bladder cancer patients. Cox proportional hazards models were used for the association of cancer subtypes to various outcomes, in both univariable (UVA) and multivariable (MVA) manners. Pearson correlation coefficients were calculated for examining the significance of correlations. Additionally, logistic regression was utilized to check the association between clusters (generated via agglomerative hierarchical clustering) and response. Again, R was selected for the execution of statistical analyses.

Kim et al. [12] used R to develop univariable and multivariable logistic regression models as well as Wilcoxon’s test to assess the predictive capacity of the Decipher test [13] to predict adverse pathology among prostate cancer patients. An interesting point to mention here is the way the similarities of the endpoints of interest were treated between the work mentioned above and this work. The adverse pathology, an indicator roughly ranging from pT1 up to pT5, was dichotomized and transformed into a binary variable. The same strategy was followed based on the patients’ Gleason score.

In many cases, there is a synergy between machine learning models and statistical tests. Kamoun et al. [14] has made an effort to achieve an international consensus pertaining to muscle-invasive bladder cancer (MIBC) subtype classification following previously proposed classifications as consistently as possible. Their proposed classifier is an ensemble of six preexisting classifiers. For the evaluation of the resulting classes with regards to clinical factors, they used many statistical methods such as Kaplan–Meier curves for the visualization of overall survival, Cox models for deriving hazard ratios (HR) for different indicators, and tests such as Fisher’s exact test for measuring associations, or Kruskal–Wallis, t-tests, and Wald tests for assessing differences between consensus classes.

Tibshirani et al. [15] first applied nearest-centroid classification in 2002 for the classification of cancer types/tumors. The method has since found multiple applications in the field, such as, for example, in Lindskrog et al. [16], where they used a Pearson nearest-centroid classifier for the identification of prognostic subtypes of non-invasive bladder cancer (NMIBC), based on the work of Kamoun et al. for MIBC. They followed an unsupervised approach through the utilization of the ConsensusClusterPlus R package (also used in this work) for the initial class generation. For differential gene expression across classes, they used analysis of variance (ANOVA). Lastly, based on the most representative genes for each class, they calculated genomic signature scores, which constitutes yet another similarity with this work.

Use of ML models, namely, Support Vector Machine (SVM), Random Forest (RF), K-Nearest Neighbors (K-NNs), and Extreme Gradient Boosting (XGB) are commonly found in cancer predictions. S. Prusty et al. [17] proposed a stratified k-fold cross-validation on the ML models mentioned above to predict cervical cancer. Two well-organized reviews have been done on this related domain. The first one was presented by Jones et al. [18] on AL and ML algorithms for early-stage detection of skin cancer in the community and primary care settings. The second one was presented by Aveta et al. [19] on different urological cancers to predict the urinary Micro-RNAs as potential biomarkers. Not only in cervical or skin cancer, AL can be utilized in liver cancers as well to find biomarkers in clinical diagnosis or primary and metastases levels [20]. Organ-wise, prostates in men are situated close to the bladder. Therefore, some of the works focused on prostate cancer at initial biopsy have been presented by Margolis in [21,22] to show the clinical performance of ExoDX (EPI) Prostate Intelliscore test. Moreover, the existing research shows the essentiality of research to find out the most impacting biomarkers in bladder cancer as well. Therefore, in this paper we have proposed an approach to address the research gaps.

## 3. Materials and Methods

The main focus of this work is the discovery of genomic biomarkers, for various clinical endpoints, among bladder carcinoma (BLCA) patients. The endpoints of interest include response to neoadjuvant chemotherapy (NAC), disease progression, survival outcome (either as time-to-progression or time-to-last followup since treatment), and T-Stage at the time of surgery. In this section, we have presented the materials (e.g., dataset, grouping technique applied for therapy response and disease progression) and methods, such as differential gene expression analysis, Kaplan–Meier estimator, dimensionality reduction, clustering, signature score generation, sample selection, and application of machine learning algorithms. All the steps used in this work are explained in detail in this section.

### 3.1. Dataset

The discovery dataset used was provided by the University of Calgary’s Cumming School of Medicine. It consists of a retrospective cohort of 102 BLCA patients who underwent chemotherapy. The clinical file, provided in the Excel workbook format, included a wide variety of indicators which are described on a high level, ranging from smoking habits and hazard exposure, to blood work and tumor stage and to survival status and various outcome dates.

Initially, the cohort contained 209 samples. However, it had to be limited to only patients coming from the University of Calgary hospital, since both the clinical and the gene expression data for patients with IDs outside the University of Calgary cohort were not available. Those patients were identified by their Diagnostic TURBT Path which was a dedicated field in the patient record that would start with an “SR-” prefix.

There are many ways to discover genomic biomarkers for disease groups. In the context of this research, the chosen approach was to analyze gene expression levels which were found in a file consisting of 46,050 unique gene expression values for 124 patients provided in .rds format. The genomic data sequencing was carried out by Decipher Biosciences [23]. Gene expression levels were normalized using the SCAN [24] algorithm.

In order to correctly match the gene expression information to the clinical data for each patient, an additional utility metadata file was needed. This was needed as well since the samples had to be limited to the intersection of those present in both the clinical and the gene expression files. By taking advantage of 2 columns in the file, the Diagnostic TURBT Path # (also found as external.id) and the CELfile name, the desired joint dataset was created.

### 3.2. Grouping Based on Therapy Response and Disease Progression

For some clinical endpoints of interest, it was deemed beneficial to lessen the degree of resolution. Specifically, the Therapy Response column consisted of 4 distinct values: 0 = Partial Response (PR), 1 = Complete Response (CR), 2 = Stable Disease (SD), and 3 = Progressive Disease (PD). Instead of looking at each category individually, the *Response* patients (PR + CR) and the *Disease* patients (SD + PD) were grouped together, respectively. Based on that, a new column was created in the clinical file mentioned above, taking the value 0 for samples in the former group, indicating the lack of the event of disease, and 1 for the latter.

A similar approach was followed in the case of the histology indicator. Information about the samples’ histology was split into 2 columns, predominant and secondary. Out of 102 samples, 98 were predominantly urothelial, with only 4 being neuroendocrine. However, information about the secondary histology was a lot more variable, taking any (or a combination) of the following values of secondary histologies: 1 = Urothelial; 2 = Squamous; 3 = Neuroendocrine/Small Cell; 4 = Plasmacytoid; 5 = Adeno; 6 = Micropapillary; 7 = Papillary; and 8 = Other.

In addition to the above, 41 samples had a missing value in this field, which signified that only the predominant histology was significant or present. Consequently, with secondary histology being the main discriminator, these 8 aforementioned categories were reduced to 5 groups, based on the process described in Algorithm 1.

Since this process ignores insignificant histological differences with respect to the problem in focus, it provides a greater number of samples for each cancer subtype group. This grouping may therefore result in more insightful analyses (e.g., survival) with respect to the clinical image of each patient, as well as potentially reveal valuable discriminating factors, such as differentially expressed genes, among the different groups.

Similarly, another clinical endpoint that underwent the same pre-processing was the T-Stage at surgery. According to the American Joint Committee on Cancer (AJCC) Staging Manual [25] there are 12 different bladder cancer tumor stages, as can be seen in Table 1.
cancers-15-04801-t001_Table 1Table 1Bladder cancer tumor stages and stage grouping.T-StageDescriptionStaging GroupDescriptionTanon-invasive papillary tumorstage 0Ta or TisTisin situ (non-invasive flat)stage IT1T1through lamina propria into sub-epithelial connective tissuesstage IIT2(a or b)T2into muscularis propriastage IIIT3(a or b) or T4aT2aonly invades inner half of the musclestage IVT4bT2binvades into outer half of the muscle

T3invasion into perivesical tissues

T3amicroscopic extravesical invasion

T3bmacroscopic extravesical invasion

T4direct invasion into adjacent structures

T4aprostate, uterus, vaginal vault

T4bpelvic side wall and/or abdominal wall


**Algorithm 1:** Histology-based sample grouping
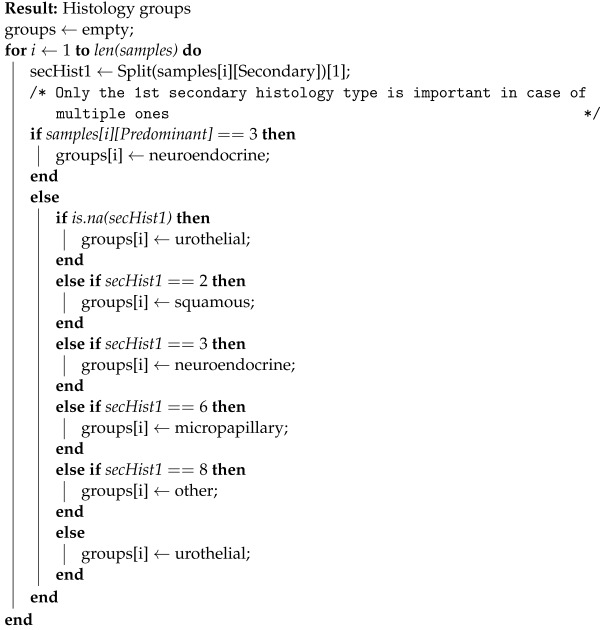


### 3.3. Differential Gene Expression Analysis

Differential gene expression (DGE) analysis is a very important step and one of the most common applications of RNA sequencing data. The main purpose of this process is to highlight potential systematic differences between the expression levels of certain genes among different patient groups. These groups may be defined by the straightforward distinction between control patients and patients who represent actual disease cases. At the same time, they can consist of sets of patients separated based on population demographics such as age or sex, health risk factors such as smoking or dietary habits, or different clinical characteristics such as cancerous tumor stage or therapy response, etc. The set of genes that might emerge as differentially expressed can then be utilized as biomarkers for either diagnostic or prognostic purposes in the clinical environment. In the scope of this work, a rather simplistic approach was followed in the direction of DGE analysis. This was mainly driven by the following two facts: the available RNA-Seq data were pre-processed (SCAN normalized) and the DE genes were not directly reported/relied upon.

A decision was made to not incorporate any of the popular packages already found in the literature (DESeq2) [26], edgeR [27,28], EBSeq [29]), but instead to simply investigate the genes available among the predefined groups for statistically significant differences in the levels of their expression. The method of choice here was the Wilcoxon rank-sum test (also known as the Mann–Whitney U test) [30]. This nonparametric statistic is a popular choice for bioinformatics applications [31] and it is a suitable one for the nature of the data available, since the expression levels were continuous values from independent samples and, although normalized, there were notable outliers. Specifically, the wilcoxTest function from the GSA-Lightning R module [32] was used in the experimental process, since its results are intuitive and it has a high performance in terms of execution time, even for a large number of candidate genes. The wilcoxTest functionality was incorporated as a custom function that can be found in the code under the name DEv2. This short pipeline (DEv2) was initiated by filtering out the majority of the genes around the mean level of expression, as explained in Section 3.5. This step made sure that wilcoxTest will be provided with a reduced number of candidate genes, which would speed up the DE analysis process further by ruling out redundancies. The last step would be the final reporting of the significantly overexpressed or underexpressed genes found for the groups of interest, based on a predefined *p*-value cutoff, also known as α.
(1)selected<−rownames(result[which(result[,1]<=p|result[,2]<=p),])

Here, *p* is the cutoff, and the *result* is a structure that holds the *p*-values for each gene, which are given by the row names. Columns 1 and 2 pertain to the over or underexpression of the genes.

#### Wilcoxon Rank Sum Test

As noted above, the Wilcoxon Rank Sum Test, also known as Mann-Whitney U test, is a very popular nonparametric method employed for the comparison of outcomes between two groups. Often described as the “nonparametric version of a two-sample *t*-test ”, it requires that the groups under comparison be independent. This requirement is always satisfied during the experimental processes followed in this work.

The purpose of such tests is to be able to reject a null hypothesis, in other words the assumption that there is no difference between two populations, with a level of certainty higher than a specific threshold (usually α for the *p*-values). Contrary to the *t*-test’s null hypothesis of equal means, the Wilcoxon Rank Sum Test’s null hypothesis assumes equal medians, indicating a focus of the latter on the whole distribution of values in each group.

### 3.4. Kaplan–Meier Estimator

As mentioned in the background section, one of the most frequently used indicators in medical research is the time elapsed from the beginning of surveillance until an event (usually death) takes place for patients belonging to a group *A* (e.g., treated), as opposed to other patients, member of a different group *B* (e.g., not treated). This amount of time is known as survival time [33], whereas the process of analyzing different patient groups for this purpose is called survival analysis.

The most popular approach to the survival analysis problem is the Kaplan–Meier (KM) estimator [34]. It is a nonparametric method which has the significant advantage that it can take into consideration right-censored data. Such is the case when a patient ceases participation in a study (stopped follow-up), or the event of interest had not occurred up to the time of the last follow-up. Unlike other methods such as the Life Table estimator [35,36], it does not require fixed time intervals and does not assume uniform censorship. A limitation is that the spectrum of its capabilities is narrow, since it can only take into account a single characteristic of the study group at hand (univariate analysis).

Also known as the Product–Limit (PL) estimate, the KM estimate can be defined in the following manner: For *N* observed lifetimes (time - to - event), assuming an ascending ordering with respect to their length, we have $0\leq t_{1}\prime \leq t_{2}\prime \leq \cdots \leq t_{N}\prime$. Then,
$$\hat{P}\left(t\right)=\prod_{r}\left[\left(N-r\right)/\left(N-r+1\right)\right]$$
where *r* indicates these values for which $t_{r}\prime \leq t$ and $t_{r}\prime$ measure the time to event. The result is a distribution which maximizes the likelihood of the observations.

For the vast majority of use-cases, the results are presented and examined in a visual manner through the Kaplan–Meier curves. These curves are easy to interpret in that they are intuitive plots that take the shape of declining horizontal steps, depicting the percentage of survival on the *y*-axis and time on the *x*-axis. The entire aforementioned process is part of the background of many popular packages of statistical software, such as the survival R package [37,38], which was utilized in the scope of this work. An additional invaluable feature of the KM method and these statistical packages is the provision of the log-rank test [39], which can assess whether the difference between the survival probability distributions of two or more independent groups is statistically significant.

### 3.5. Dimensionality Reduction and Clustering

As mentioned in the dataset section, the discovery cohort used include gene expression levels for 46,050 genes. This is a very large set of variables, which greatly outnumbers the number of available samples and contains a lot of uninformative data. Not all genes that are profiled are overexpressed or inhibited in a population of cancer patients, hence seeking potential biomarkers among the entire group adds unnecessary delays due to the complexity of the discovery process which increases the run-time without providing useful insights.

A simple yet effective additional step for commencing the differential expression analysis on a limited but more suitable candidate genes subset is to select a mean expression level cutoff. A cutoff value of 0.7 was selected in order to discard a significant part of genes irrelevant to a cancerous condition. This value also allows for greatly inhibited genes to be included.

It is important to note that the starting set of genes had been initially reduced to the intersection of the ones present in the UofC cohort by the ones found in data generated by the TCGA Research Network [40], accessed through cBioPortal [41,42,43]. The Cancer Genome Atlas (TCGA) is considered a landmark cancer genomics program, providing the necessary genomic data to a plethora of similar research works [44]. Therefore, limiting the number of genes was accordingly deemed reasonable, not only for dimensionality reduction purposes, but also as a step towards further compatibility with external validation sets. The Figure 1 shows the gene expression level range for a sample of 12,000 genes.

As mentioned in the “Logistic Regression” subsection, a crucial part of the dimensionality reduction pipeline is the usage of the elastic-net regularization method for the logistic regression model. This model behaves particularly well in sparse modeling problems, such as the one studied here.

Another technique employed for the same purpose of selecting a few suitable features out of a deluge of candidate variables is the popular Principal Component Analysis (PCA) [45,46,47]. This method, which is used to change the basis of the data at hand, is particularly useful when trying to visualize clustering results. In particular, a multi-feature clustering result can easily be plotted on a Cartesian plane after reducing the number of dimensions (features) down to two principal components.

In brief, the PCA consists of the following steps:1.Data standardization: This is a very important first step to ensure that each continuous variable included in the analysis process has an equal contribution (this is a crucial step for clustering as well). Variables with significantly larger ranges (variances) than other ones might dominate the resulting Principal Components, since they will seemingly be carrying more information. The most simple standardization formula is
$$z=\frac{value-mean}{standard\;deviation}$$
where *z* is the scaled data point, and it is implemented in Python’s sklearn library as the sklearn.preprocessing.StandardScaler class [48].2.The next step is the calculation of the covariance matrix. This information will allow for the discovery of potentially correlated variables and, therefore, the elimination of redundancies.3.Finally, in order to identify the Principal Components, the eigenvectors and eigenvalues of the covariance matrix have to be calculated. Sorting the eigenvalues in a descending order essentially reveals the ordering of the most important vectors, meaning the ones that explain the greatest amount of variance of the data.

An early effort to cluster the patients incorporated the above process through a Python pipeline. Specifically, sets of genes (signatures) that would be found as potentially correlated to a certain clinical endpoint were used for variable selection. Then, following the data scaling, the clustering process would take place. Arguably the most popular clustering method, *K-Means* is available through the sklearn package as the sklearn.cluster.KMeans class, and it was the clustering method of choice for this research. In order to come up with the optimal number of consistent clusters, the silhouette method [49] and the knee/elbow heuristic [50] were used. Finally, using the matplotlib module’s functions, the clusters formed could be clearly visualized. An example of this can be seen in Figure 2.

Unfortunately, the resulting clusters, although they usually managed to distinguish well most of the therapy responders from the non-responders, did not provide adequate results for robust conclusions in the clinical setting, according to a medical expert. For that reason, the rest of the clustering analysis took place in the R language pipeline, based on different methods as well as different packages.

In the field of bioinformatics, specifically in the genomic data analysis area, clustering is heavily used, not always as a standalone method but often in conjunction with heatmaps [51,52,53].

For the experimental processes in this research, clustering was implemented through the ConsensusClusterPlus R package [54]. This module is based on the technique of consensus clustering [55]. The idea behind this approach is to use the consensus of multiple runs of a clustering algorithm in order to assess the stability of the discovered clusters. This powerful module provides a range of clustering methods. The one chosen here was the Partitioning Around Medoids (PAM) [56].

PAM clustering is more robust compared to the more popular K-means approach, since it tries to minimize a number of dissimilarities instead of squared Euclidean distances of each data point to a central cluster point. PAM, however, is noted to be susceptible to the issue of handling large or high-dimensional datasets, therefore making it less desirable for clustering gene expression data. However, this does not pose an issue in this work since the high dimensionality issue is being addressed prior to the clustering step, and the number of genes has been greatly reduced.

Aside from the algorithm, another important aspect of the clustering process is the distance metric. Unlike the aforementioned Euclidean distances or any geometrically defined metric, Spearman correlation [57] values were used. Grouping genes together based on the correlation of their level of expression is of the utmost importance in this context, and could bring to light relationships that could result in forming entire gene signatures. Being a nonparametric approach, Spearman’s correlation also accounts for the skewness in the distribution of the gene expression values.

Visualization of clustering results was achieved through the utilization of the heatmap3 R package [58].

The heatmap, initially appearing as a shading matrix to aid in statistical analysis [59,60], has become an almost indispensable tool for bioinformatics analysis pipelines.

This data visualization technique provides a compact description of the quantitative variables of interest in a 2D matrix format containing area marks colored with a diverging colormap. In its cluster form, the heatmap includes trees drawn on the borders showing how the matrix is ordered with respect to the derived clusters which, in turn, are the result of the hierarchical clustering applied on the data. Specifically, in the field of computational biology, the depicted data involve the level of expression of a set of genes along a group of different samples (patients). For example, in Figure 3, one can find the examined genes on the horizontal axis (columns) and the various samples on the vertical axis (rows). Assuming that green is chosen to signify overexpression, we can draw the conclusion that the leftmost genes are overexpressed between the 6 top samples. Likewise, darker shades would hint at a level of expression closer to 0, and red shades would indicate gene inhibition, in a similar manner pertaining to color intensity indicating level of expression.

### 3.6. Signature Score

In order to be able to use a potentially meaningful signature in conjunction with time-to-event data under an estimator, such as the Kaplan–Meier, the nature of the predictor variables had to change. To be specific, a predictive model based on a certain gene signature would utilize the level of gene expression for each one of the genes that comprise the signature itself. These levels are, by definition, continuous variables. A discretization step is therefore necessary at this point.

A first step towards a straightforward solution to this problem is to create a signature score. Adopted by other research work in the field such as Hsiao et al. [62], this approach essentially allows for a single value to be generated for each sample in the dataset, with respect to an endpoint of interest. Since most of the clinical endpoints considered were of binary nature, the developed, underlying prediction model would be a binary logistic regression one. By default, such a model produces as output the probability of a certain sample belonging in either the event-positive class (usually 1, to signify event presence) or the event-negative class (usually 0, to signify the absence of the event). This value, called the response in R’s glmnet models, is rather suitable for achieving the discretization goal, since it nicely quantifies the chances that a sample exhibits an event or not.

The essential discretization process happens as a second step, where the aforementioned signature scores are grouped into different risk categories. Since the score itself reflect the probability of the event of interest taking place or not, the splitting does not have to be very sophisticated. For that purpose, the quartile base R function was employed. The output of said function is the a set of threshold values that define the quartiles of the data at hand. Quartiles divide the data points in four parts in an orderly manner, such that the first quartile is the middle value between the minimum value available and the median, the second quartile is the median itself, and the third quartile is the value between the median and the data maximum [63,64]. Once this process is completed, a single discriminating trait (the set in which a sample belongs) can be used to split the samples in, e.g., 2 risk groups (low and high), which in turn lays the foundation for plotting the Kaplan–Meier curves.

### 3.7. Sample Selection

One of the main difficulties in this work was obtaining two separate patient groups, one for discovery purposes and the other for validation. The solution employed here was to split the data into training and test sets, as is traditionally done in machine learning model training. A common practice in many research works that involve biomarker discovery or the analysis of different risk groups in general also includes using an “in-house” patient cohort for discovery and external groups for validating potential findings. Given the peculiarities of the University of Calgary cohort, meaning that it consists of urothelial bladder cancer patients that underwent neoadjuvant chemotherapy, it is particularly demanding to find a suitable validation set.

One such dataset available online was deposited to the European Genome-Phenome Archive [65,66], under accession number EGAS#00001002556. There were mainly two issues with utilizing this patient cohort. First, the bureaucratic process required in order to get access to the data was significantly lengthier than initially expected, with the first inquiry taking place on 31 May 2021, and the data only becoming available on 1 August 2021, allowing for little to no time to work with it. In addition to that, as opposed to the format of data utilized in this work, which is gene expression levels as continuous variables, the RNA sequencing information provided through this dataset was in unprocessed FASTQ [67] format, containing raw read counts. Not only would this require a copious amount of pre-processing in order to generate the gene expression level values, which is outside the scope of this work, but also the uncompressed size of it exceeded 1 TB, which was prohibitive for the computational resources available.

After extensive research of candidate validation sets and experimenting with some of them, such as a collection of samples curated by multiple TCGA platforms [43], a decision was made to rely on internal validation. This process, although it might hinder the capacity of any results for wide generalization in the bladder cancer patient population, ensures that no conclusions are drawn based on improper comparisons (e.g., treated vs. untreated patients when researching therapy response).

One of the major issues that had to be mitigated was the lack of fixed training and test sets during model development. Each time a logistic regression model was created, new patients would be picked for training and validation. As mentioned previously, stratified sampling was ensured; however, there was no guarantee that the composition of the sets would remain identical. Furthermore, a consequent implication of this is the fact that based on the patients the model would be trained on, the genes that would be picked as model predictor variables would change as well. To tackle these problems initially, internal validation was approached through a model “consensus”.

The four main components of the pipeline are, in order, the DEv2 custom function for the simple differential expression analysis step, the training of an elastic-net regularized logistic regression model, the counter of the selected gene occurrences, and the final logistic regression model (again, following elastic-net regularization).

During the first step, the execution of DEv2, a simple filtering was applied in order to eliminate candidate genes where the level of expression was very close to 0. The threshold value was set to 0.7. Following that, a Mann–Whitney U test was conducted, with the significance threshold *alpha* taking a value of 0.01, which can be considered more strict, significantly reducing the number of genes qualifying as differentially expressed.

Having reduced the potential predictor genes from tens of thousands to usually at most a few hundred, the second step was to create a logistic regression model. The important point here, however, is not the model, but the selection of predictor variables. The elastic-net regularization process eliminated most genes and was limited to just a few that were optimal for said model.

These genes were counted in the occurrence counter and which concluded one gene picking cycle. This process was repeated 100 times. After its completion, the most frequently picked genes were gathered in a vector, in descending order of frequency.

The 100 most frequent genes were, in their turn, fed as the input for the final logistic regression model creation, where the elastic-net once again picked the best predictors for the model. This method focused on “standardizing” the genes, and not the patients, for the training set. An abstract illustration of the pipeline is provided in Figure 4.

A second, more complex but also more complete approach was developed, having as a main objective the standardization of the patient sets. The key idea was picking a training-testing split that achieved the maximum overlap (Figure 5) with a group of previously selected genes.

The first stage for this approach was an iterative process, highly similar to the consensus approach. There was, however, a significant difference: the gene selection is better informed since the entire dataset is taken into consideration for the model creation. The frequency of the selected genes is not as important as the size of the overlap; therefore, any gene that is selected even once is kept in a gene collection. Once the collection is complete, the process goes into a second, very similar iterative phase, where the differentially expressed genes are more scrutinized (α=0.01 vs 0.05 previously), and the models are created with a 70/30 train/test ratio. The reason for the latter is that the accuracy of the models will be taken into consideration in order to achieve the best overlap/accuracy trade-off. The created models are collected through 100 cycles in a list structure, along with their training sets, their test sets, their predictor variables (genes), and their prediction accuracies.

The final step is the execution of the find_overlap function. This module compares the overlap of each model’s genes with the initial gene collection and picks the model (and consequently its train/test sets) as the best, only if the gene overlap is bigger than the previously found and the accuracy of the new model is not worse than the one already acquired. The output of this function led to fixing the training set for the final model creation, and in doing so it greatly impacted the gene selection for the resulting signature.

### 3.8. Machine Learning Implementation

One of the main objectives for discovering a biomarker related to a certain clinical endpoint is, ultimately, to be able to build a model that will ideally have prognostic capabilities. Since many such endpoints indicate either the occurrence or the absence of an event X∈{0,1}, they are inherently binary in nature. Hence, a reasonable and evidently popular method used is logistic regression [68].

Binary logistic regression utilizes the logistic function by log-transforming the odds ratio. The result of this step is called a *logit* and it is defined as follows:$$logit\left(P\right)=\ln\frac{P}{1-P}$$
*P* is the result of the logistic function, also known as the *sigmoid function*
$$P=f\left(x\right)=\frac{1}{1+\epsilon^{-x}}=\frac{\epsilon^{x}}{\epsilon^{x}+1}$$
and it reflects the probability of a sample belonging in a certain class (0 or 1). Lastly, *x* can be defined as a linear regression equation
$$x=a+b_{1}x_{1}+b_{2}x_{2}+\ldots +b_{N}x_{N}$$
where *a* is the constant, also called *intercept* of the equation, and $b_{i}$ is the coefficient of predictor variable *i*, with i∈[1,N].

In order to be able to classify a sample *x*, we need a decision boundary. For this kind of applications a boundary of 0.5 is usually picked, such that
$$\mathrm{class}\left(x\right)=\left\{\begin{array}{cc} 1 & \mathrm{if}\;P(y=1|x)>0.5\\ 0 & \mathrm{otherwise} \end{array}\right.$$

The exact method that was chosen was the *elastic-net* regularized regression [69]. The elastic- net is a linear combination of the *LASSO* [70] and *ridge* [71,72] methods, an approach that overcomes the potential non-uniqueness of the LASSO [73]. This problem arises when the number of variables *P* exceeds the number of samples *N*, a scenario that is rather commonplace in genomic data analysis.

The methods above were implemented with the use of the R package glmnet [74]. Initially, the discovery cohort (UofC dataset) was split into training and test sets following a 70/30 ratio. The splitting was carried out by the partition function of the R package splitTools [75]. The default behavior of this module is *stratified* splitting, ensuring that an adequate number of samples from all classes will be present in each set. Following that step, cv.glmnet was employed in order to complete a *k*-fold cross-validation (*k* set to 10) and come up with the optimal regularization parameter λ (lambda.min) that minimizes the mean cross-validation error. Finally, the glmnet model training function was called, with α=0.5 to set the type of the model to elastic-net and the λ parameter set to the previously discovered lambda.min (equivalent to calling glmnet.fit with the appropriate λ).

## 4. Results and Discussion

As mentioned in Section 3, two of the main endpoints of interest for the bladder cancer patients were Disease Progression and Therapy Response. In order to create a predictive model for these two, the Maximum Overlap process (Figure 5) was followed. After coming up with a fixed composition for the training and the test sets, upon which the logistic regression models would be trained and evaluated, a pool of candidate genes was also picked (as they were produced from the same process). After the final, elastic-net regularized model was created, the following results were obtained for therapy response and disease progression.

### 4.1. Qualified Gene Signatures

#### 4.1.1. Therapy Response

The qualified gene signatures are: ALG1L, ARL17A, CEP164, DNAJC5G, FBXO36, HIST1H3I, HMBS, KCNS3, NHP2L1, NR2C2AP, PCDHB6, PDZD11, RPL9, RPSAP52, S100A3, SLC44A5, SRY, SYCE2, ZNF287, ZNF584, MECR, ABO, ACOT2, ADRB3, C5orf60, CUZD1, FAM24B, FAM84B, GABRA3, GABRQ, GLYATL3, NUDT22, OASL, PFAS, PROM2, PVRIG, RAPGEF4, SIAH2, SNAPIN, TTC12, SIRPD, CHST1, TRPV5, PTPN18, DCAF12L2, PARP16, NOP2, BNIP3, ABCC2, LMBRD2, MRPL55, SMG6, CMTM4, TMEM141. The Accuracy obtained with 95% CI 0.589–0.9041 is 77.42%. The Size of overlapped genes is 55 and the AUC obtained is 0.7174. Figure 6 shows the AUC ROC for predicting therapy response.

#### 4.1.2. Disease Progression

The qualified genes are: ACBD7, ACOT2, ATHL1, CLDN6, CX3CR1, DCAF12L2, DNAJC15, ELL3, FCGR1A, FITM2, GDI1, GEMIN8P4, HBA2, HIST1H3D, HIST1H4E, HMBS, ID4, IQCC, LRP2BP, MAGEE2, MOCOS, NLGN4Y, NUDT8, PLA2G4E, PVT1, RHOXF1, RPL9, RPS6KA6, SLC25A4, SSTR4, TMEM11, TTTY15, VMAC, ZCCHC4, BTNL2, POM121L10P, DAPK2, SLC24A3, KCNMB1, SEMA3D, SPACA4, DCDC2B, ZNF213, ZFY, C1orf198, CCDC159, KLHL13, SERPINE3, SGPP2, TMEM200A, ACTL7B, CCNF, WDR47, ARRDC1. The accuracy obtained with 95% CI 0.7862–0.9834 using the logistic regression model is 92.11%. The Size of overlapped genes is 57 and the AUC obtained is 1.0. Figure 7 shows the Area Under the ROC curve for predicting Disease Progression.

The above result is interesting because it essentially means that one has obtained the perfect predictor, or in other words the classes (event of progression vs. no event) are perfectly separated. In order to potentially further improve the predictor’s performance, R’s cutpointr module was once again utilized.

Finding the optimal class separation threshold (cutpoint) was done for both endpoints. For Therapy Response, that value was found to be equal to 0.9535. Note that this value refers to each model’s response, which is the probability of each sample for a certain class membership. Considering the newly found threshold, the accuracy improves by approximately 3%, reaching 80.7%. For Disease Progression, however, the improvements are even greater. The reason for such an improvement can be somewhat implied by the rather high 95% Confidence Intervals reported above. With a revised threshold drawn at a probability of 0.21%, the model is a perfect predictor, correctly classifying all samples.

This is a particularly interesting result, since after a meticulous effort to eliminate any type of selection bias, two gene signatures emerged, capable of predicting either eight out of 10 times or perfectly the outcome of very important indicators for cancer patients.

#### 4.1.3. Time-to-Event Prediction Results

Unrelated to genomic biomarkers per se, additional insights were gained with regards to the correlation of various clinical indicators and the potential for some of them to act in a predictive manner.

The first noteworthy result is related to the prediction of time-to-disease progression, based on the grouping of Therapy Response groups. With a *p*-value = 0.0023, indicating a very significant correlation, said grouping can be a trustworthy estimator of the amount of days it will take for certain percentages of patients to exhibit disease progression after NAC (Figure 8).

Following a similar rationale, the indicator of T-stage at surgery was found to be significantly correlated with estimating the time-to-disease progression (*p*-value = 0.0086). See Figure 9.

Lastly, another very insightful result was also obtained which pertained to the grouping of bladder carcinoma subtypes (mostly primarily urothelial) The generated sets of patients could act as a significantly correlated estimator (*p*-value = 0.0053) for the same endpoint of time-to-disease progression (Figure 10).

The power of such findings lies in the fact that clinicians can presumably improve or ascertain their decision making, may that be for a treatment plan or for the duration of watchful waiting [76], without the need for genomic profiling, but simply through considering a patient’s clinical image and electronic health records.

#### 4.1.4. Patients Clustering

The last reported result in this section is the outcome of a heatmap provided with the PAM algorithm’s clustering results of the patients as input. All patients and the most frequent genes as they occurred from the Consensus method (see Figure 4) were considered. Aside from the level of gene expression, there is a superposition of additional information. This includes most of the crucial clinical indicators, as well as the generated signature scores and the various group memberships (histology & therapy response). The result can be seen in Figure 11. Abundantly informative, even though it might be chaotic, there are multiple conclusions that can be drawn already through a brief visual inspection of the emerging patterns. First of all, there is a significant alignment of Partial and Complete Therapy Response with T-Stage at surgery < T2. The orange and pink clusters combined contain only a handful of therapy responders, and their included patients show a very distinctive overexpression of the majority of the selected genes (seen in red on the wide bottom left corner). Conversely, on its majority, the black cluster exhibits an underexpression of those same genes. Moreover, the patients that constitute roughly the left 40th percentile of the black cluster show an underexpression of approximately the first 65 genes on the right side of the figure. These patients, although they do not show Disease Progression in the relevant indicator, demonstrate a state of Progressive Disease under Therapy Response. This subset of individuals might be of great interest for further investigation, assuming that their disease status has changed over time, but only after the administration of NAC.

## 5. Conclusions

In this paper, various recent works were explored related to bladder cancer biomarkers. Different researchers have used a plethora of approaches to this topic from classic statistical approaches to very elaborate, state-of-the-art machine learning methods including Deep Networks, to different ones stemming from the perspective of systems biology, such as pathway analysis. We have focused on grouping based on therapy response and disease progression on the patients. Using the histology-based sample grouping, gene expression analysis, Wilcoxon Rank Sum Test, and clustering, we have reported some of the biomarkers which have exceptionally high correlation with the bladder cancer patient cohort. The Kaplan–Meier curves are presented for the survival analysis of the patients with explanation. One of the challenges we have faced which can be considered for future development is external validation. With the support of external datasets, we may validate the hypothesis more accurately.

## Figures and Tables

**Figure 1 cancers-15-04801-f001:**
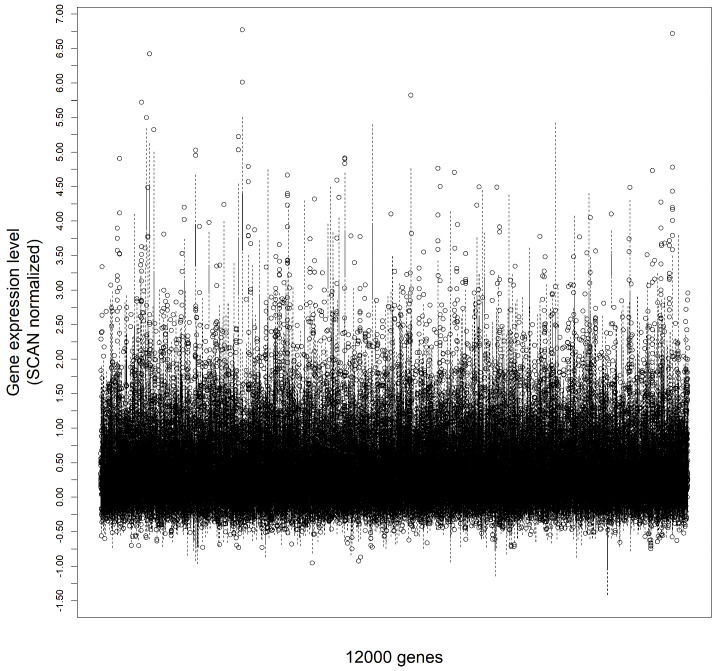
Gene expression level range for a sample of 12,000 genes.

**Figure 2 cancers-15-04801-f002:**
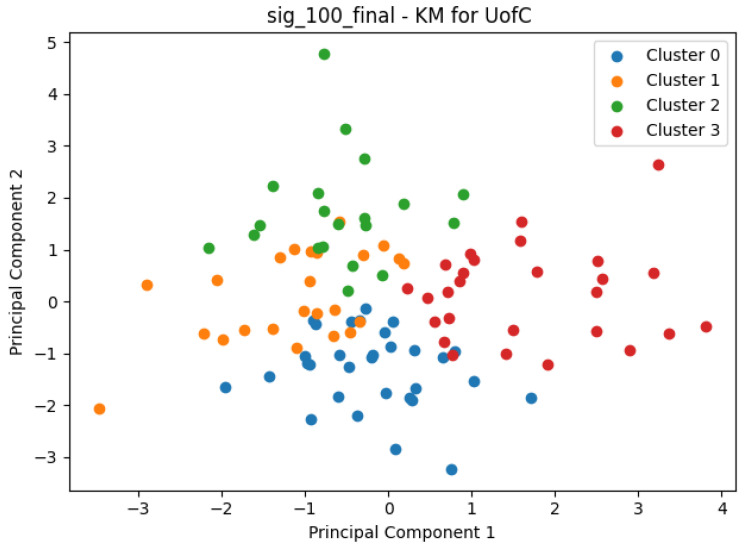
K-Means clustering, with K=4, for all patients available in the UofC dataset, based on the expression of the following genes: ATHL1, RPL9, HMBS, ARL15, OASL, CX3CR1, ID4, DCAF12L2, FAH, LRTM1, TTLL9.

**Figure 3 cancers-15-04801-f003:**
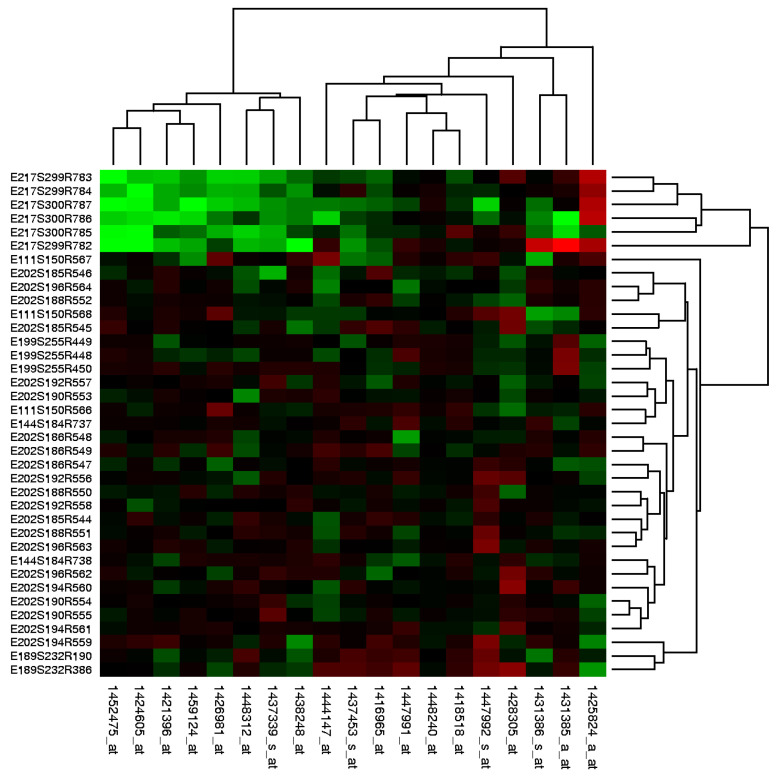
Heat map generated from DNA microarray data reflecting gene expression values in several conditions [61].

**Figure 4 cancers-15-04801-f004:**
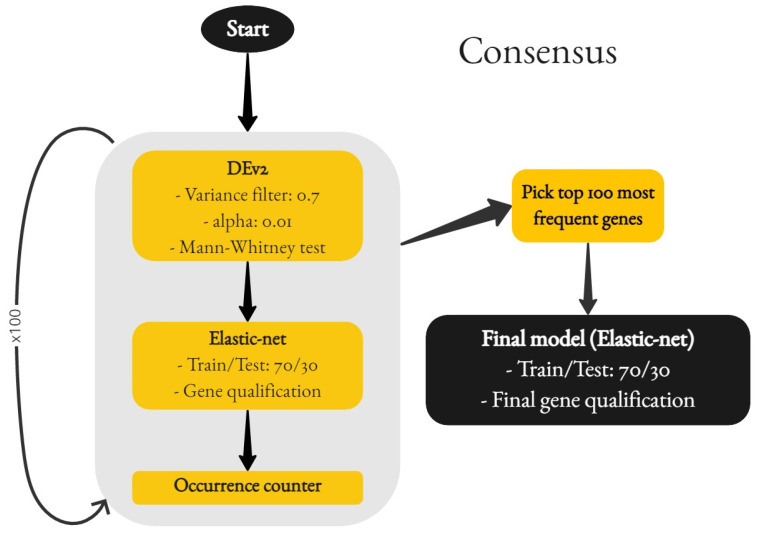
Consensus gene selection.

**Figure 5 cancers-15-04801-f005:**
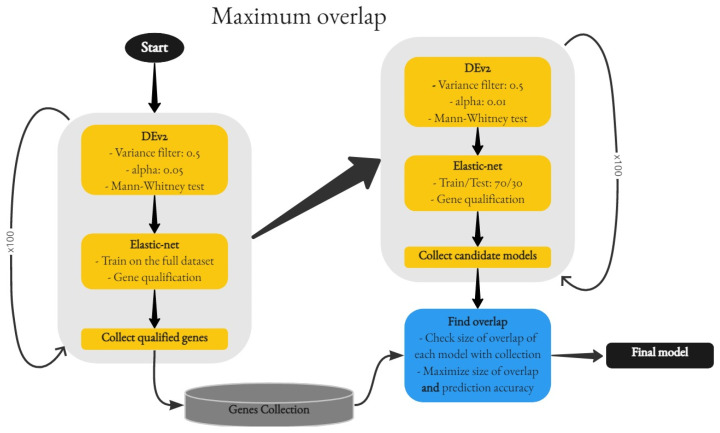
Maximum overlap sample and gene selection.

**Figure 6 cancers-15-04801-f006:**
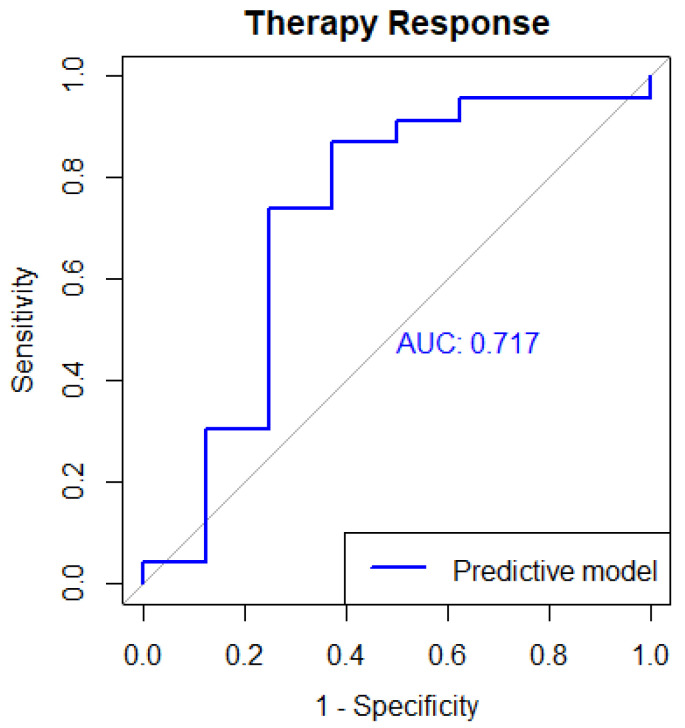
Area Under the ROC curve for predicting Therapy Response.

**Figure 7 cancers-15-04801-f007:**
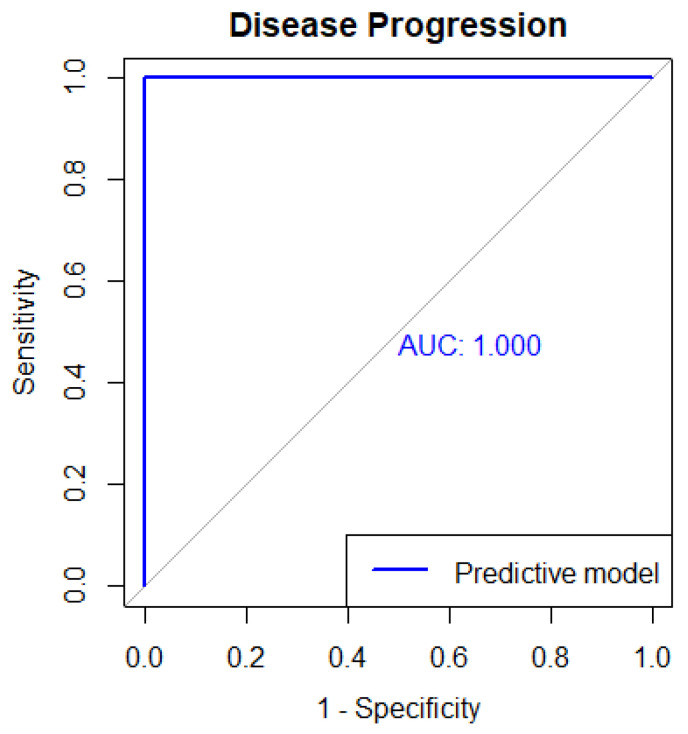
Area Under the ROC curve for predicting Disease Progression.

**Figure 8 cancers-15-04801-f008:**
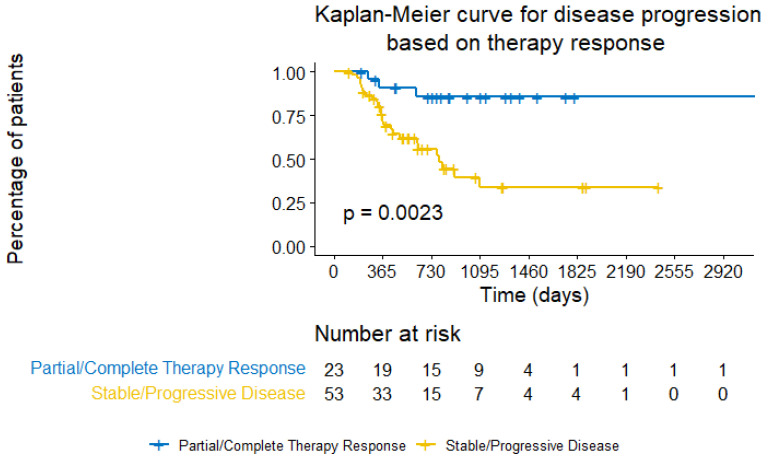
Kaplan–Meier curve for disease progression based on therapy response.

**Figure 9 cancers-15-04801-f009:**
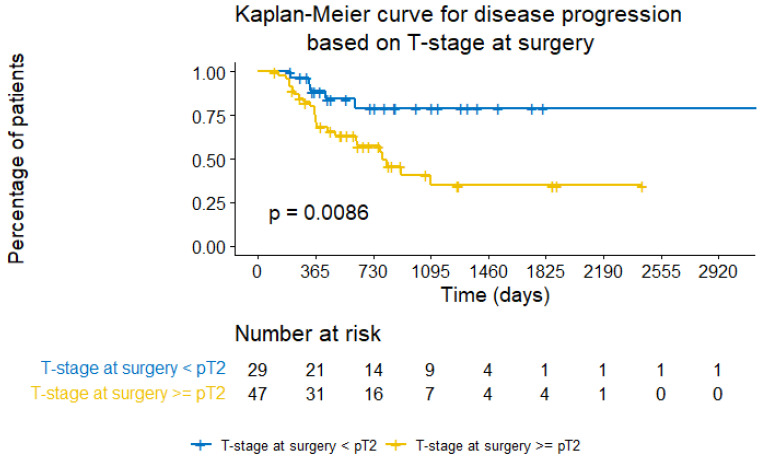
Kaplan–Meier curve for disease progression based on T-stage at surgery.

**Figure 10 cancers-15-04801-f010:**
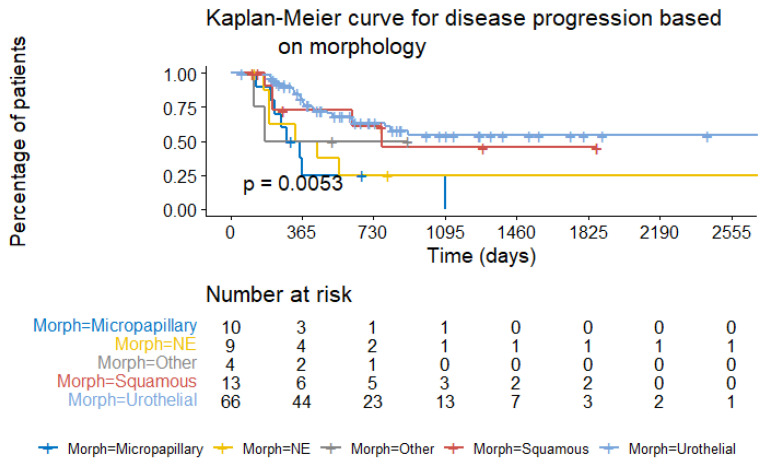
Kaplan–Meier curve for disease progression based on tumor morphology.

**Figure 11 cancers-15-04801-f011:**
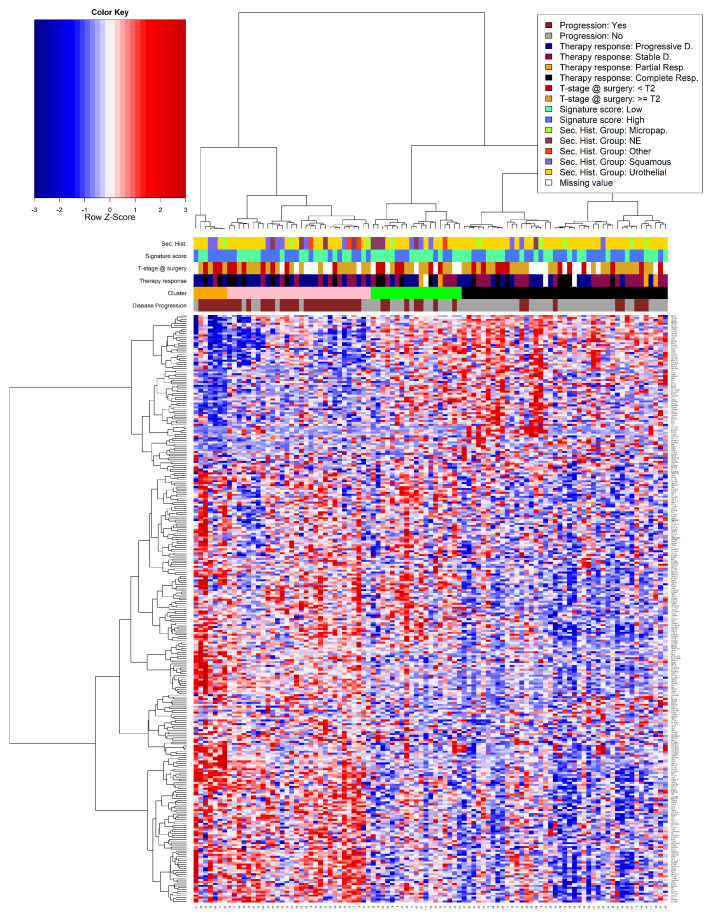
Heatmap of clustering results.

## Data Availability

Data are available from the corresponding author and can be shared with anyone upon reasonable request.

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
