# Peer review of "Genomic Biomarker Discovery in Disease Progression and Therapy Response in Bladder Cancer Utilizing Machine Learning"

_cancers, 2023, doi:10.3390/cancers15194801_

Round 1

Reviewer 1 Report

The authors have nicely presented a manuscript using machine learning for the discovery of genomic biomarkers for bladder cancer. The manuscript is clear and easy to understand. Overall, this is a very important field of research and requires urgent attention hence the objective of this manuscript is relevant and recent. Although the manuscript is impressive, I have a few minor revisions before it can be accepted for publication:

1. There are a few grammatical mistakes and sentence formation errors in the manuscript that need to be corrected. Please proofread the manuscript before submitting the revision.

2. Although the authors focus on bladder cancer, it is important to discuss overall progress in biomarker discovery through ML and DL tools and how any previous literature can be used to guide the current research in bladder cancer. For this, the authors could also prepare a summarized table comparing previous studies and how various methods can be reproduced for bladder cancer biomarker discovery. Additionally, while discussing the recent literature in the manuscript, please discuss and cite the following important papers:

- Prusty, Sashikanta, Srikanta Patnaik, and Sujit Kumar Dash. "SKCV: Stratified K-fold cross-validation on ML classifiers for predicting cervical cancer." Frontiers in Nanotechnology 4 (2022): 972421.

  - Jones, O. T., et al. "Artificial intelligence and machine learning algorithms for early detection of skin cancer in community and primary care settings: a systematic review." The Lancet Digital Health 4.6 (2022): e466-e476.   - Bakrania, Anita, et al. "Artificial intelligence in liver cancers: Decoding the impact of machine learning models in clinical diagnosis of primary liver cancers and liver cancer metastases." Pharmacological Research 189 (2023): 106706.   - Margolis, Erik, et al. "Predicting high-grade prostate cancer at initial biopsy: clinical performance of the ExoDx (EPI) Prostate Intelliscore test in three independent prospective studies." Prostate cancer and prostatic diseases 25.2 (2022): 296-301.   The above mentioned literature can be included in the "relevant works" section. Please also change the sub-heading "relevant works" to a more specific sub-heading that more specifically defines the section for readers.

Overall, minor English editing is required. There are a few grammatical errors in the manuscript that need to be proof-read before re-submission.

Author Response

Thank you so much for your effort and time to review this paper. We appreciate your contribution and guidance for the paper. The detail responses has been added in the attached file based on your valuable comments.

Reviewer 2 Report

the manuscript is interesting in the aim. however, needs a deep improvement.

  • The whole introduction needs to be restructured and would benefit from a clearer statement of the study's aims and objectives, less general information about the study aim, and the inclusion of more recent literature to contextualize the current state of knowledge and the knowledge gap the study seeks to fill. Do not be redundant and I strongly suggest shortening.
  • Figure 1 needs to be eliminated. 
  • Section “2. Related works” should be eliminated and only the data considered essential may be included in the discussion.
  • Authors should read these novel papers to update their references and find a new point of discussion PMID: 37446024.
  • Check typos

Author Response

Thank you so much for your effort and time to review this paper. We appreciate your contribution and guidance for the paper. We have revised the paper according to your valuable comments. 

Round 2

Reviewer 2 Report

The manuscript has undergone enhancements. In my view, it possesses merit for being published.